# RETRIEVAL-AUGMENTED META TEST-TIME TRAINING FOR MULTIMODAL REASONING

## ABSTRACT

Reasoning lies at the core of Large Vision–Language Models (LVLMs). Recent Test-Time Scaling (TTS) methods enhance reasoning by allocating additional computation during inference. However, they primarily exploit the model's internal knowledge without incorporating new information, which limits their effectiveness under distribution shifts. While retrieval can introduce new knowledge, existing methods primarily emphasize semantic similarity rather than reasoning utility, leaving LVLMs struggle to effectively leverage the retrieved examples for complex reasoning. To address these limitations, we propose **RAM-TTT** (**R**etrieval-**A**ugmented **M**eta **T**est-**T**ime **T**raining), a retrieve–train–generate framework that unifies retrieval with meta-adaptation. RAM-TTT includes two key components: (1) LVLM Aligned Retrieval (LAR), which selects examples for both semantic relevance and reasoning utility, and (2) Meta Test-Time Training (Meta TTT), which casts retrieved examples as alternating support sets and meta-queries, allowing the model to "learn how to learn" from retrieved examples while mitigating overfitting. Experiments show consistent gains on MathVerse (+6.4%), LogicVista (+5.6%), and We-Math (+8.5%) with Qwen2-VL-7B, and strong generalization to Phi-3.5-Vision and Pixtral-12B. These results highlight RAM-TTT's broad applicability in enabling LVLMs to acquire and internalize new information at test time for stronger reasoning under distribution shifts.

## 1 INTRODUCTION

Reasoning lies at the core of Large Vision–Language Models (LVLMs) (Wang et al., 2024b; Chen et al., 2024; Li et al., 2024), underpinning tasks such as mathematical problem solving (Zhang et al., 2024; Lu et al., 2023) and logical reasoning (Xiao et al., 2024). Recent advances in Test-Time Scaling (TTS) (Snell et al., 2024; Muennighoff et al., 2025) show that allocating more computation at inference can noticeably enhance reasoning, for instance by sampling multiple reasoning trajectories (Wang et al., 2022) or through iterative refinement (Madaan et al., 2023). However, such approaches often falter under out-of-distribution (OOD) scenarios (Caffagni et al., 2024), as they recycle the same internal knowledge without introducing new information, leaving the model's reasoning under-activated (Hu et al., 2024). This limitation motivates our goal of equipping test-time scaling with the ability to incorporate external reasoning signals, enabling more reliable adaptation under distribution shifts.

A straightforward approach is Retrieval-Augmented Generation (RAG) (Lewis et al., 2020; Caffagni et al., 2024), which can introduce new knowledge during inference and thus partially alleviates this limitation. Yet existing methods are typically optimized for semantic similarity, overlooking whether the retrieved examples actually facilitate deeper reasoning. Consequently, these examples may appear relevant to the test query but contribute little to genuine reasoning.

Furthermore, LVLMs often struggle to effectively adapt and internalize retrieved examples, even when useful examples are provided (Baldassini et al., 2024; Shukor et al., 2023), underscoring the necessity of adaptation mechanisms. Test-Time Training (TTT) (Sun et al., 2020) fulfills this role by updating model parameters at inference using query-specific signals (typically self-supervision) derived from the test query (refer to Figure 1 (a)). It has shown considerable promise for encoder-based (Shu et al., 2022; Yoon et al., 2024; Kojima et al., 2025) and sequence models (Gandelsman et al., 2022; Sun et al., 2024). However, directly fine-tuning an LVLM on a handful of retrieved examples is prone

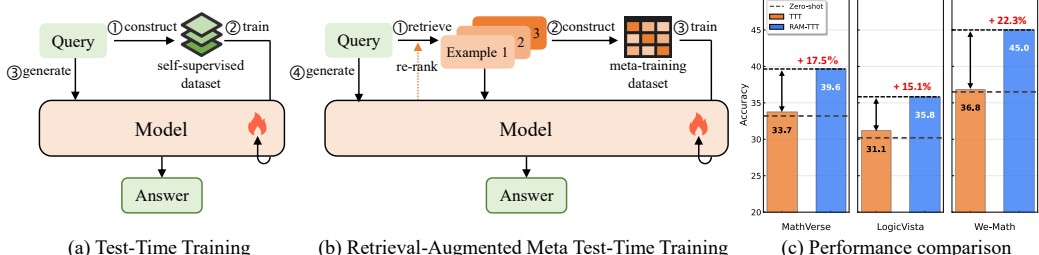

(a) Test-Time Training     (b) Retrieval-Augmented Meta Test-Time Training     (c) Performance comparison

Figure 1: **Comparison of TTT, and our proposed RAM-TTT.** (a) TTT updates the model based on the query, often using self-supervision, without leveraging extra examples in model training. (b) Our proposed RAM-TTT first retrieves examples according to the query, re-ranks them with LVLM, and then constructs a meta-training dataset for test-time training. After the model is trained, the answer to the query is generated with the retrieved examples as context. (c) RAM-TTT consistently outperforms TTT across benchmarks (MathVerse, LogicVista, We-Math).

to overfitting and poor generalization to the test query. Thus, an effective test-time paradigm must couple retrieval with robust adaptation, enabling LVLMs to fully exploit retrieved examples without sacrificing generalization.

To address these challenges, we propose **RAM-TTT** (**R**etrieval-**A**ugmented **M**eta **T**est-**T**ime **T**raining), a novel retrieve–train–generate framework that unifies retrieval with meta-adaptation. RAM-TTT is guided by two principles: (1) retrieval should capture not only semantic relevance but also alignment with the LVLM's reasoning process; and (2) adaptation should avoid overfitting on limited supervision while still enabling the model to internalize new information at test time.

Building on these principles, we develop two key components. First, **LVLM Aligned Retrieval** (LAR) selects examples not only for semantic relevance but also for their contribution to the LVLM's reasoning, ensuring that the retrieved examples can enhance reasoning. Second, inspired by meta-training (Min et al., 2021; Sinha et al., 2024), **Meta Test-Time Training** (Meta TTT) introduces a meta-learning perspective: retrieved examples alternately act as support sets and meta-queries in the In-Context Learning (ICL) format. This design mitigates overfitting and reduces adaptation difficulty under limited supervision, allowing the LVLM to "learn how to learn" from retrieved examples. Together, these components integrate retrieval with adaptation, enabling LVLMs to both acquire and internalize new information for stronger reasoning under distribution shifts. Extensive experiments demonstrate strong and consistent improvements: +6.4% on MathVerse, +5.6% on LogicVista, and +8.5% on We-Math with Qwen2-VL-7B. Moreover, our experiments reveal that RAM-TTT generalize well to other LVLMs such as Phi-3.5-Vision and Pixtral-12B, thereby demonstrating its broad applicability across architectures and benchmarks.

Our main contributions are summarized follows: **(1)** We propose RAM-TTT, a unified retrieve–train–generate framework that equips LVLMs to acquire and internalize new information at test time for stronger reasoning. **(2)** We design the LAR module to select examples not only by semantic relevance but also by their utility for the model's reasoning. **(3)** We develop a Meta TTT paradigm, where retrieved examples alternately serve as support sets and meta-queries in the ICL format, effectively mitigating overfitting and improving adaptation.

## 2 METHODOLOGY

In this work, we propose **R**etrieval-**A**ugmented **M**eta **T**est-**T**ime **T**raining (RAM-TTT), a novel retriever-trainer-generator framework for improving LVLM performance on complex reasoning tasks at test time. As illustrated in Figure 2, RAM-TTT integrates two key components: (1) an **LVLM Aligned Retrieval** (LAR) module, which uses both CLIP and LVLM confidence to retrieve examples not only by semantic relevance but also by their contribution to the LVLM's reasoning. (2) a **Meta Test-Time Training** (Meta TTT) paradigm, which casts retrieved examples as alternating support sets and meta-queries in the ICL format, allowing the LVLM to "learn how to learn" from retrieved

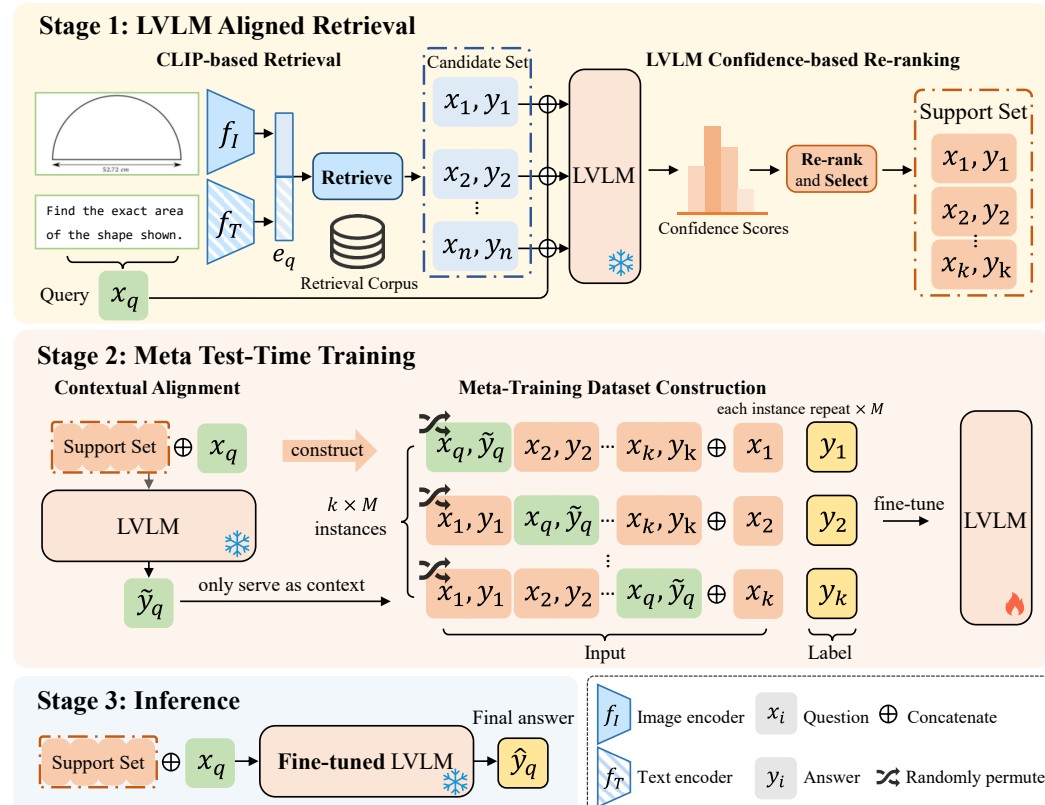

Figure 2: **Overview of our RAM-TTT framework.** The proposed framework contains two novel components: (1) **LVLM Aligned Retrieval (LAR):** given a query, retrieve relevant and LVLM-compatible examples from the external corpus; (2) **Meta Test-Time Training (Meta TTT):** adapt the LVLM to the query using contextual alignment and meta-training on the support set. Once the LVLM is adapted, we use it to generate the final answer for the query.

examples while mitigating overfitting. After this meta-training step, the adapted model is used to generate the final prediction for the query.

## 2.1 PROBLEM SETUP

Multimodal reasoning tasks, such as mathematical problem solving (Lu et al., 2023; Zhang et al., 2024; Wang et al., 2024a) and logical reasoning (Xiao et al., 2024), typically combine visual and textual information in the query and require sophisticated reasoning to arrive at the correct answer. Formally, let a query be $x_q = (I_q, T_q)$, where $I_q$ is the input image and $T_q$ is the textual question. We have access to a retrieval corpus $\mathcal{C} = \{(x_i, y_i)\}_{i=1}^N$ consisting of $N$ examples, where each example is a question-answer pair $(x_i, y_i)$. Here, $x_i = (I_i, T_i)$ consists of an image $I_i$, a textual question $T_i$, and $y_i$ is its corresponding answer with intermediate reasoning steps. The primary goal is to predict the answer $y_q$ for a given query $x_q$ with an LVLM by leveraging relevant examples from the corpus $\mathcal{C}$ at test time. Formally, this can be expressed as:

$$y_q = \text{LVLM}(\mathbf{I}_{\text{std}}, \mathcal{S}, x_q), \tag{1}$$

where $\mathcal{S}$ is the support set of relevant examples retrieved from $\mathcal{C}$ for the query $x_q$. Here, $\mathbf{I}_{\text{std}}$ refers to the standard ICL instruction prompt, which guides the model to learn from the provided examples and generate both the reasoning process and the final answer for the query. The detailed prompt of $\mathbf{I}_{\text{std}}$ is provided in Appendix B.

## 2.2 LVLM ALIGNED RETRIEVAL

Given an input query $x_q = (I_q, T_q)$, LAR is proposed to select a small support set $\mathcal{S}$ from the retrieval corpus $\mathcal{C}$ according to $x_q$. Specifically, LAR operates in two sequential stages: CLIP-based candidate retrieval and LVLM confidence-based re-ranking.

**Stage 1: CLIP-based Retrieval.** In this stage, we first identify a set of semantically relevant candidates. For the query $x_q$ and each example $x_i = (I_i, T_i)$ in the corpus $\mathcal{C}$, we use CLIP (Radford et al., 2021) encoders ($f_I, f_T$) to extract image and text embeddings and concatenate them to form representations $e_q$. We then compute the cosine similarity $\text{Sim}(x_q, x_i)$ between the query and each example. Based on these scores, we select the top-$m$ most similar examples to form a candidate set $\mathcal{C}_{\text{cand}}$. We set $m = r \times k$, where $k$ is the final desired support set size and $r$ is a retrieval ratio hyperparameter. After this stage, the examples in the candidate set exhibit high semantic relevance to the query.

**Stage 2: LVLM Confidence-based Re-ranking.** In this stage, we re-rank candidates in $\mathcal{C}_{\text{cand}}$ based on the LVLM confidence score. For each candidate $(x_i, y_i)$, we build an in-context learning (ICL) prompt $P_i = [\mathbf{I}_{\text{dir}}, x_i, y_i, x_q]$, where $\mathbf{I}_{\text{dir}}$ is a direct-answer instruction (see Appendix B). The LVLM then produces a tentative answer $\hat{y}_q^{(i)} = \text{LVLM}(P_i)$.

We measure the LVLM's confidence in $\hat{y}_q^{(i)}$ using the inverse perplexity. Perplexity, defined as the exponential of the average negative log-likelihood, is a standard indicator of model uncertainty (Radford et al., 2019), and has been widely used as a proxy for model confidence in recent work on LLM calibration (Geng et al., 2024). The confidence score is

$$\text{Conf}_q^{(i)} = \frac{1}{\text{PPL}_q^{(i)}}. \tag{2}$$

Intuitively, a lower perplexity means the LVLM internally considers the candidate's reasoning pattern more helpful, implying higher reasoning utility without requiring any additional training or calibration. We select the top-$k$ candidates with the highest confidence scores to form the final support set $\mathcal{S}$.

## 2.3 META TEST-TIME TRAINING

After obtaining the high-quality support set, we introduce the Meta TTT module to further adapt the LVLM to the query $x_q$ by leveraging the retrieved examples through a meta-training objective. We first propose a novel contextual alignment strategy to augment the support set to better align training and inference contexts, and then construct an effective meta-training dataset to fine-tune LVLM.

**Contextual Alignment Strategy.** To better align the LVLM with the specific query $x_q$, we first construct a contextual alignment instance $\tilde{y}_q$ for $x_q$ by prompting the LVLM with the standard instruction $\mathbf{I}_{\text{std}}$ and the support set $\mathcal{S}$. We then augment the support set with the contextual alignment instance: $\mathcal{S}' = \mathcal{S} \cup \{(x_q, \tilde{y}_q)\}$. This reduces train–test mismatch by ensuring that the query appears in both training and inference contexts.

**Meta-Training Dataset Construction.** Our meta-training process uses a leave-one-out strategy over the original support set $\mathcal{S}$. For each ground-truth example $(x_i, y_i) \in \mathcal{S}$, we treat it as the prediction target for a training step. The context for this step is composed of the remaining $k-1$ support examples along with the contextual alignment instance. Crucially, the contextual alignment instance $(x_q, \tilde{y}_q)$ serves only as an ICL example; model parameters are updated exclusively based on the loss computed on the ground-truth pairs from $\mathcal{S}$. To enhance diversity and robustness to context order, we generate $M$ random permutations of the context for each target, denoted as $\text{Context}_i^{(l)}$ for $l = 1, \ldots, M$. Each meta-training prompt is constructed as:

$$P_{\text{meta}}^{(i,l)} = [\mathbf{I}_{\text{std}}, \text{Context}_i^{(l)}, x_i]. \tag{3}$$

The complete meta-training dataset $\mathcal{D}_{\text{meta}}$ is the collection of all such prompt-answer pairs:

$$\mathcal{D}_{\text{meta}} = \left\{ \left( P_{\text{meta}}^{(i,l)}, y_i \right) \mid (x_i, y_i) \in \mathcal{S}, \; l = 1, \ldots, M \right\}. \tag{4}$$

**Training Loss.** We adapt the LVLM parameters by minimizing the following loss:

$$\mathcal{L}_{\text{TTT}} = - \sum_{(P,y) \in \mathcal{D}_{\text{meta}}} \sum_{t=1}^{|y|} \log p(y_t \mid P, y_{<t}). \tag{5}$$

## 2.4 INFERENCE

After Meta TTT, the adapted LVLM with parameters $\theta'$ is used to generate the final answer. The inference prompt, $P_{\text{test}}$, is constructed using the standard instruction $\mathbf{I}_{\text{std}}$, the original support set $\mathcal{S}$, and the query $x_q$. The adapted model then predicts the final answer:

$$\hat{y}_q = \text{LVLM}_{\theta'}(P_{\text{test}}). \tag{6}$$

## 3 EXPERIMENTS

### 3.1 EXPERIMENTAL SETUP

**Benchmarks.** We evaluate our proposed RAM-TTT on four widely-used multimodal reasoning benchmarks: MathVerse (Zhang et al., 2024), MathVista (Lu et al., 2023), LogicVista (Xiao et al., 2024), and We-Math (Qiao et al., 2024). Concretely, MathVerse, MathVista, and We-Math primarily assess visual mathematical reasoning, covering diverse subjects, diagram understanding, and knowledge generalization, while LogicVista focuses on logical reasoning in visual contexts. In particular, for MathVerse, we exclude the Vision-Only split and report the experimental results on the remaining splits.

**Retrieval Corpus Construction.** To construct a diverse multimodal retrieval database for our proposed RAM-TTT, we process and aggregate MultiMath-300K (Peng et al., 2024), LLaVA-CoT-100k (Xu et al., 2024), and Geo170K (Gao et al., 2023). For LLaVA-CoT-100k, we retain only the reasoning and conclusion segments. For Geo170K, we keep only entries with chain-of-thought (CoT) annotations and remove duplicates that have the same image. All samples are reformatted into a unified structure: <think> *reasoning process* <\think> Answer: *final answer*, explicitly separating reasoning and answer to help the LVLM utilize retrieved examples during adaptation. The final multimodal retrieval corpus contains 403,210 examples. Further data processing details are provided in Appendix C.

**Backbone and Baselines.** We adopt Qwen2-VL-7B (Wang et al., 2024b) as the LVLM backbone and use CLIP ViT-L/14@336px (Radford et al., 2021) for retrieval. We compare our RAM-TTT with the following baselines:

**(1) Zero-shot:** direct inference with the pretrained LVLM, without adaptation or context.
**(2) Self-Correction:** test-time iterative refinement of predictions (Madaan et al., 2023).
**(3) TTT:** parameter-efficient fine-tuning on CLIP-retrieved examples with supervised objectives.
**(4) ICL:** direct inference with the pretrained LVLM by leveraging $k$ CLIP-retrieved support examples as context, without parameter updates (Shukor et al., 2023).

**Hyperparameters and Implementation Details.** We use only 4-shot in-context examples in all main experiments, but explore all few-shot settings in the ablation study. Hyperparameters and implementation details are provided in Appendix A. The source code will be released soon.

### 3.2 MAIN RESULTS

As summarized in Tables 1 and 2, our **RAM-TTT** consistently outperforms all baselines across four benchmarks with respect to overall metrics, demonstrating the effectiveness of both the LAR and Meta TTT modules. The experimental analysis for each benchmark is detailed below.

On **MathVerse**, our RAM-TTT surpasses Zero-shot and TTT by 6.4% and 5.9% gains in average accuracy, respectively. The most significant gains are observed on the Text-dominant (+10.2%) and Text-lite (+7.3%) splits, indicating that LAR effectively identifies support examples that are not only relevant but also beneficial for the LVLM's reasoning process. Our RAM-TTT also maintains robust gains on vision-intensive splits, highlighting its ability to generalize across different modalities.

On **LogicVista**, according to overall accuracy, our RAM-TTT outperforms Zero-shot and ICL by 5.6% and 2.5% gains in overall accuracy, respectively. Notably, the improvements in Deductive and Numerical reasoning categories suggest that Meta Test-Time Training enables the model to better adapt to diverse logical structures, beyond what is achieved by standard ICL.

On **We-Math**, our RAM-TTT achieves an 8.5% increase over Zero-shot and a 3.9% gain over ICL with respect to overall accuracy. The reduction in Insufficient Knowledge (IK) errors and the

Table 1: **Comparative results on MathVerse and LogicVista.** MathVerse: TD (Text Dominant), TL (Text Lite), VI (Vision Intensive), VD (Vision Dominant); LogicVista: Ded (Deductive), Ind (Inductive), Mec (Mechanical), Num (Numerical), Spa (Spatial), ALL(overall accuracy).

| Method | MathVerse | | | | | LogicVista | | | | | |
|---|---|---|---|---|---|---|---|---|---|---|---|
| | TD↑ | TL↑ | VI↑ | VD↑ | AVG↑ | Ded↑ | Ind↑ | Mec ↑ | Num↑ | Spa↑ | ALL↑ |
| Zero-shot | 37.3 | 32.4 | 31.3 | 31.6 | 33.2 | 51.6 | 27.1 | 24.3 | 26.3 | 19.2 | 30.2 |
| Self-Correction | 42.4 | 35.3 | 31.1 | 32.0 | 35.2 | 53.8 | 22.4 | 36.5 | 27.4 | **26.9** | 33.1 |
| TTT (4-shots) | 36.9 | 32.9 | 32.2 | 32.7 | 33.7 | 50.5 | **28.0** | 25.7 | 29.5 | 19.2 | 31.1 |
| ICL (4-shots) | 42.0 | 35.5 | 33.2 | 33.6 | 36.1 | 54.8 | 24.3 | **44.6** | 28.4 | 15.4 | 33.3 |
| **RAM-TTT (4-shots)** | **47.5** | **39.7** | **36.8** | **34.4** | **39.6** | **55.9** | **28.0** | **44.6** | **30.5** | 20.5 | **35.8** |

Table 2: **Comparative results on We-Math and MathVista.** We-Math: IK (Insufficient Knowledge), IG (Inadequate Generalization), CM (Complete Mastery), RM (Rote Memorization); MathVista: TQA (Textbook Question Answering), VQA (Visual Question Answering), GPS (Geometry Problem Solving), MWP (Math Word Problem), FQA (Figure Question Answering), ALL(overall accuracy).

| Method | We-Math | | | | | MathVista | | | | | |
|---|---|---|---|---|---|---|---|---|---|---|---|
| | IK↓ | IG↓ | CM↑ | RM↓ | Overall ↑ | TQA↑ | VQA↑ | GPS | MWP↑ | FQA↑ | ALL↑ |
| Zero-shot | 56.8 | 5.5 | 33.7 | 10.6 | 36.5 | 60.8 | **55.3** | 40.9 | 65.1 | 65.1 | 57.6 |
| Self-Correction | 48.8 | 12.2 | 34.9 | 10.7 | 41.0 | 57.6 | 42.5 | 45.7 | 60.2 | 58.4 | 53.1 |
| TTT (4-shots) | 57.0 | **5.3** | 34.1 | 9.6 | 36.8 | 61.4 | 54.7 | 42.3 | 65.1 | **66.5** | 58.3 |
| ICL (4-shots) | 50.5 | 10.1 | 36.0 | **8.7** | 41.1 | 60.8 | 51.4 | 45.2 | 60.2 | **66.5** | 57.3 |
| **RAM-TTT (4-shots)** | **44.6** | 11.1 | **39.4** | 11.2 | **45.0** | **63.9** | 53.1 | **52.4** | 66.1 | 63.9 | **60.0** |

increase in Complete Mastery (CM) further demonstrate RAM-TTT's capacity to facilitate knowledge acquisition and generalization in complex multi-step reasoning scenarios.

On **MathVista**, Self-Correction shows a marked drop in overall performance, highlighting the limitation of relying solely on the model itself without external knowledge. ICL also underperforms Zero-shot, particularly on TQA and MWP splits, where limited relevance between test questions and the retrieval corpus exposes the weakness of similarity-based retrieval. In contrast, our RAM-TTT achieves 2.4% gain in overall accuracy over Zero-shot. This can be attributed to the effectiveness of LAR: by selecting support examples that are both relevant to the query and aligned with the LVLM's reasoning, LAR enables the model to leverage useful reasoning strategies from the retrieval corpus, even when direct matches are lacking. Combined with Meta TTT, this allows RAM-TTT to achieve robust test-time adaptation and generalization to tasks with limited retrieval relevance.

Collectively, the above main results demonstrate that RAM-TTT's integration of LAR and Meta TTT facilitates robust and generalizable improvements for complex multimodal reasoning tasks, establishing strong performance for test-time adaptation in LVLMs.

### 3.3 ABLATION STUDY

**Component Analysis.** In Table 3, we conduct an incremental ablation study, starting from the basic TTT and progressively adding each core component of RAM-TTT: Meta TTT, confidence-based re-ranking, and contextual alignment. Our findings are as follows: (1) **+ Meta TTT.** Adding the Meta TTT module leads to a clear improvement over the basic TTT. This is because standard TTT fine-tunes on the entire support set as a single batch and easily overfits to a few retrieved examples, while Meta TTT alternates each example as a meta-query and uses the rest as temporary support via leave-one-out construction. This episodic structure forces updates to generalize across retrieved examples rather than memorizing any single instance, substantially reducing single-example overfitting and improving adaptation stability. (2) **+ Confidence Re-ranking.** Further incorporating confidence-based re-ranking brings additional gains, especially on MathVerse (from 36.6% to 38.0%). This highlights the benefit of selecting examples that better align with LVLM, thereby further enhancing the LVLM's reasoning capability. (3) **+ Contextual Alignment.** Finally, incorporating the contextual alignment strategy yields the highest overall performance. This step provides a significant boost (e.g. from 38.0% to 39.6% on MathVerse), which confirms that including the query with a contextual alignment

Table 3: **Ablation study** for the main components of RAM-TTT on MathVerse and LogicVista.

| Method | MathVerse | | | | | LogicVista | | | | | |
|---|---|---|---|---|---|---|---|---|---|---|---|
| | TD↑ | TL↑ | VI↑ | VD↑ | AVG↑ | Ded↑ | Ind↑ | Mec ↑ | Num↑ | Spa↑ | ALL↑ |
| TTT | 36.9 | 32.9 | 32.2 | 32.7 | 33.7 | 50.5 | 28.0 | 25.7 | 29.5 | 19.2 | 31.1 |
| + Meta TTT | 44.4 | 35.8 | 33.8 | 32.4 | 36.6 | 54.8 | 23.4 | **44.6** | **33.7** | 16.7 | 34.5 |
| + Conf. Re-ranking | 45.7 | 38.5 | 34.5 | 33.2 | 38.0 | 54.8 | **29.0** | **44.6** | 30.5 | 17.9 | 35.3 |
| + Contextual Align | **47.5** | **39.7** | **36.8** | **34.4** | **39.6** | **55.9** | 28.0 | **44.6** | 30.5 | **20.5** | **35.8** |

Table 4: **Ablation studies** on CLIP models and similarity computation strategies.

**(a) CLIP Models**

| CLIP Models | TD↑ | TL↑ | VI↑ | VD↑ | AVG↑ |
|---|---|---|---|---|---|
| ViT-B/32 | 44.9 | 36.9 | 33.0 | 31.6 | 36.6 |
| ViT-B/16 | 45.2 | 37.2 | 34.8 | 31.7 | 37.2 |
| ViT-L/14 | 44.7 | 38.6 | 36.7 | 32.5 | 38.1 |
| **ViT-L/14@336px** | **47.5** | **39.7** | **36.8** | **34.4** | **39.6** |

**(b) Similarity Computation**

| Similarity Comput. | TD↑ | TL↑ | VI↑ | VD↑ | AVG↑ |
|---|---|---|---|---|---|
| Image Only | 41.5 | 37.3 | 34.1 | 32.9 | 36.5 |
| Text Only | 43.4 | 36.4 | 36.0 | 33.2 | 37.3 |
| Mean Similarity | 44.7 | **40.9** | **37.1** | 34.0 | 39.1 |
| **Concat** | **47.5** | 39.7 | 36.8 | **34.4** | **39.6** |

Table 5: **Ablation study** of examples permutation times $M$ on MathVerse and LogicVista.

| $M$ | MathVerse | | | | | LogicVista | | | | | |
|---|---|---|---|---|---|---|---|---|---|---|---|
| | TD↑ | TL↑ | VI↑ | VD↑ | AVG↑ | Ded↑ | Ind↑ | Mec ↑ | Num↑ | Spa↑ | ALL↑ |
| 1 | 44.9 | 39.3 | 35.0 | 33.0 | 38.1 | 52.7 | 23.4 | 44.6 | **30.5** | **24.4** | 34.7 |
| 2 | **47.5** | 39.7 | **36.8** | **34.4** | **39.6** | **55.9** | 28.0 | 44.6 | **30.5** | 20.5 | **35.8** |
| 3 | 46.3 | 40.2 | 36.0 | 33.2 | 39.0 | 52.7 | **28.0** | **45.9** | **30.5** | 21.8 | 35.6 |
| 4 | 46.2 | **40.7** | 38.1 | 32.9 | 39.5 | 50.5 | 27.1 | 44.6 | **30.5** | 19.2 | 34.2 |

instance makes the meta-training dataset more closely match the inference scenario, further enhancing test-time adaptation.

**Effects of CLIP Model and Similarity Computation Strategy.** In Table 4 (a), we investigate the impact of different CLIP models on MathVerse. We observe that larger and higher-resolution models yield consistent improvements: ViT-L/14@336px achieves the highest average accuracy (39.6%), outperforming ViT-B/32 (36.6%) and ViT-B/16 (37.2%). Stronger CLIP encoders produce more discriminative joint image–text embeddings, leading to more accurate retrieval candidates that better match the query's semantic and visual structure, which in turn strengthens downstream adaptation. In Table 4 (b), we further conduct experiments on different similarity computation strategies in the CLIP retrieval stage. We find that only using the image or text features for retrieval leads to lower accuracy ("Image Only" and "Text Only"), demonstrating that multimodal reasoning requires both modalities to be jointly considered. Compared to separately averaging similarities of the two modalities ("Mean Similarity"), concatenating the modal features ("Concat") yields better performance because it allows the retrieval to capture cross-modal interactions that are essential for diagram-intensive mathematical reasoning.

**Effects of Permutation times $M$.** Table 5 presents the impact of the number of permutation times $M$ when constructing meta-training samples. We observe that: (1) Increasing $M$ from 1 to 2 leads to a clear improvement in overall accuracy on both MathVerse and LogicVista. This indicates that moderate permutation helps the model better utilize the support set by exposing it to diverse in-context example orders, thereby enhancing its generalization and robustness. (2) However, further increasing $M$ to 3 or 4 does not yield additional gains or even slightly decreases, suggesting that excessive permutations may introduce redundancy without providing additional meaningful diversity. Overall, setting $M = 2$ is sufficient to achieve strong results while maintaining efficiency.

**Effects of Retrieval Ratio $r$.** We further analyze the impact of the retrieval ratio $r$ on model performance, as shown in Figure 3. On MathVerse, RAM-TTT achieves its highest accuracy at $r = 8$ (39.6%), while on LogicVista, the best result is at $r = 4$ (35.8%). Increasing $r$ expands the candidate pool and improves the chance of retrieving high-utility examples, but excessively large $r$ introduces semantically similar yet reasoning-irrelevant examples, adding noise that weakens the

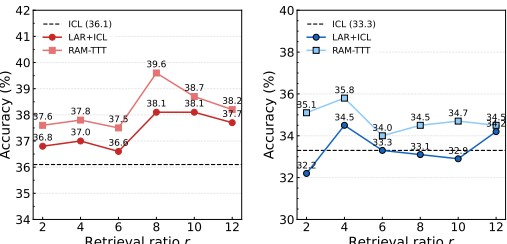
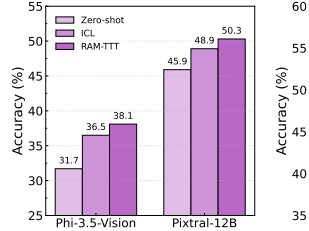
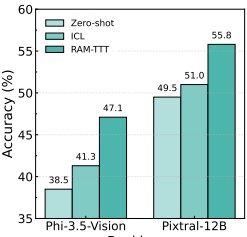

Figure 3: Effect of $r$ on model performance on MathVerse (left) and LogicVista (right).

Figure 4: RAM-TTT with different backbones on MathVerse (TD, left) and MathVista (GPS, right).

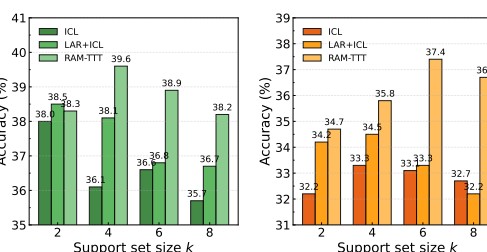
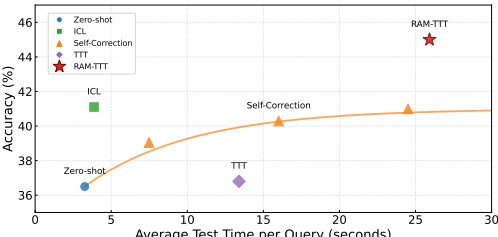

Figure 5: Sample utilization analysis with varying support set size $k$ on MathVerse (left) and LogicVista (right).

Figure 6: Analysis of accuracy–time trade-offs on We-Math across test-time strategies, conducted on one NVIDIA A800 GPU.

meta-updates. Comparing with the CLIP-based Retrieval + ICL baseline (dashed line), RAM-TTT consistently outperforms ICL across all $r$ values and datasets, demonstrating that meta-adaptation is robust even when retrieval contains mild noise. In contrast, LAR+ICL—without adaptation—remains more sensitive to retrieval noise, particularly on LogicVista.

**Generalization to Other Backbones.** To verify that our proposed RAM-TTT is a model-agnostic framework, we extend our evaluation to two additional LVLM backbones: Phi-3.5-Vision and Pixtral-12B. As shown in Figure 4, our proposed RAM-TTT consistently achieves performance improvements over both zero-shot and CLIP-based ICL baselines on MathVerse (TD) and MathVista (GPS) splits. Specifically, it raises Phi-3.5-Vision's accuracy from 31.7% (Zero-shot) and 36.5% (CLIP Retrieval + ICL) to 38.1%, and improves Pixtral-12B from 45.9% and 48.9% to 50.3%. Similar improvements are also observed on MathVista (GPS). These results confirm that RAM-TTT is broadly applicable and can effectively enhance complex reasoning across diverse LVLM backbones.

## 4 ANALYSIS AND DISCUSSION

Beyond reporting benchmark gains, we further analyze *how* RAM-TTT works by examining five aspects: (1) its ability to leverage retrieved samples, (2) its efficiency in converting additional computation into reasoning improvements, (3) its robustness to overfitting as the number of adaptation steps increases, (4) its robustness to prompt design, and (5) the effect of confidence-based example selection. These empirical findings complement our main results, while theoretical justification is deferred to Appendix E.

**Sample Utilization Analysis.** Figure 5 shows the performance of RAM-TTT and two baselines with varying support set size $k$. This reflects how efficiently different methods can leverage retrieved examples. We observe three key phenomena: (1) *Monotonic gains under moderate $k$.* When $k$ increases from 2 to 6, RAM-TTT exhibits consistent improvement, whereas baselines either plateau or degrade. This highlights RAM-TTT's ability to transform additional examples into meaningful reasoning signals rather than mere noise. (2) *Robustness under larger $k$.* When $k$ increases further to 8, all methods face accuracy drops due to noise from less relevant examples. However, RAM-TTT only shows a mild decrease, while baselines suffer steep declines. (3) These findings indicate that RAM-TTT enables a more comprehensive yet selective use of external examples, striking a balance

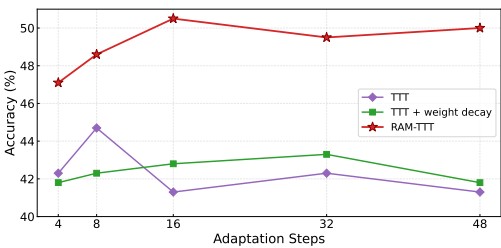 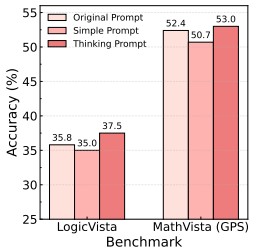 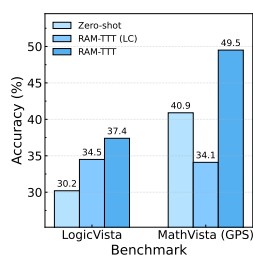

Figure 7: Adaptation Stability and Overfitting Analysis under increasing adaptation steps on MathVista (GPS).

Figure 8: Prompt robustness analysis (left) and the effect of confidence-based example selection (right).

between leveraging useful reasoning cues and filtering out irrelevant signals. Consistent patterns observed on LogicVista further highlight the generality of this phenomenon.

**Accuracy–Time Trade-off Analysis.** Figure 6 presents the trade-off between accuracy and computational time on one NVIDIA A800 GPU. The key insights are: (1) ICL benefits from retrieved examples with modest cost but cannot fully exploit their reasoning value. Self-Correction, in contrast, allocates more computation but mainly recycles internal knowledge, leading to diminishing returns. Both highlight the limitation of current test-time scaling methods without genuinely new information. (2) TTT consumes substantial computation but does not reach competitive accuracy. Direct fine-tuning on limited test signals makes it prone to overfitting, which in turn undermines its ability to generalize robustly to test queries. (3) In contrast, RAM-TTT delivers superior accuracy with the most efficient compute trade-off by channeling resources into structured meta-adaptation that integrates novel retrieved knowledge. Thus, extra FLOPs yield genuine reasoning improvements, explaining its consistent dominance on the accuracy–time frontier.

**Adaptation Stability and Overfitting Analysis.** To evaluate robustness, we plot accuracy as a function of adaptation steps for TTT, TTT with weight decay, and RAM-TTT on MathVista (GPS) (Figure 7). Three patterns emerge. (1) *Standard TTT quickly overfits.* Accuracy increases slightly in the first few steps but then degrades rapidly, showing the instability of fine-tuning on tiny support set. (2) *Weight decay offers limited mitigation.* Adding an explicit regularizer yields only marginal improvements and still exhibits clear degradation as the number of steps grows, suggesting that conventional regularization is insufficient in this setting. (3) *Meta TTT remains stable across all step ranges.* RAM-TTT improves steadily in the early steps and does not exhibit performance collapse even with many additional updates. This stability stems from its meta-learning objective, which discourages the model from overfitting to individual examples and produces adaptation that is robust to the number of adaptation steps.

**Prompt Robustness Analysis.** To assess whether RAM-TTT relies on a specific ICL template, we evaluate three types of prompts on Qwen2-VL-7B: the original prompt, a simplified prompt that only lists examples as shown in Figure 12, and a thinking prompt that additionally encourages step-by-step reasoning as shown in Figure 13. The bar plots in Figure 8 demonstrate that RAM-TTT remains stable across these variations. All prompts achieve similar performance, around 50–53% on MathVista GPS and 35–38% on LogicVista, while the thinking prompt attains the best results. These findings indicate that RAM-TTT does not depend on meticulously crafted prompts and stays robust under reasonable prompt choices once retrieved examples and meta-adaptation are incorporated.

**The Effect of Confidence-based Example Selection.** LAR ranks candidate examples using the LVLM confidence score introduced in Section 2.2 and selects the top-$k$ items as the support set. To directly assess this design, we compare RAM-TTT with high-confidence selection against a variant that deliberately uses the lowest-confidence candidates (both with $k=6$). As shown in Figure 8, low-confidence selection substantially hurts performance on MathVista (GPS), dropping to 34.1%, which is even worse than the zero-shot baseline at 40.9%, whereas high-confidence selection reaches 49.5%. On LogicVista, low-confidence achieves 34.5% compared to 37.4% for high-confidence and 30.2% for zero-shot. These results support our view that inverse perplexity is a reliable proxy for reasoning utility: high-confidence examples consistently provide useful supervision, whereas low-confidence ones behave more like noisy or adversarial signals.

## 5 RELATED WORK

**Large Vision-Language Models for Multimodal Reasoning.** Large Vision-Language Models (LVLMs) (Wang et al., 2024b; Chen et al., 2024; Li et al., 2024) have achieved strong progress but still face challenges on complex reasoning tasks such as mathematical problem solving (Lu et al., 2023; Zhang et al., 2024) and logical reasoning (Xiao et al., 2024). To enhance reasoning, prior work explores specialized fine-tuning with reasoning-focused datasets (Shi et al., 2024; Guo et al., 2024), and prompting techniques like chain-of-thought and multimodal extensions (Wei et al., 2022; Zhang et al., 2023). However, these methods often require costly retraining or heavily engineered prompts, and still struggle to generalize to diverse or out-of-distribution reasoning tasks. These challenges motivate our exploration of adaptive test-time paradigms, forming the foundation of RAM-TTT.

**Test-Time Scaling.** Recent advances in Test-Time Scaling (TTS) demonstrate that allocating additional computation during inference can substantially improve reasoning performance without retraining (Snell et al., 2024; Muennighoff et al., 2025). Self-Consistency (Wang et al., 2022) enhances robustness by aggregating multiple reasoning chains, while Tree-of-Thoughts (Yao et al., 2023) expands the search space through structured exploration with lookahead and backtracking. Self-Correction (Madaan et al., 2023) further strengthen outcomes by enabling models to revise and improve upon initial solutions. However these methods primarily recycle existing knowledge, limiting their effectiveness under distribution shifts, which our framework addresses by coupling retrieval with adaptation.

**Retrieval-Augmented In-Context Learning.** In-Context Learning (ICL) enables models to improve task performance by conditioning on retrieved examples (Brown et al., 2020; Lewis et al., 2020; Chen et al., 2022). By incorporating explicit reasoning steps in examples, ICL can boost models' reasoning abilities (Wei et al., 2022). However, similarity-based retrieval (e.g., CLIP-based) is often misaligned with LVLM reasoning needs, and LVLMs struggle to exploit complex multimodal contexts without specialized adaptation (Baldassini et al., 2024; Shukor et al., 2023; Doveh et al., 2024). This motivates our design of LVLM-Aligned Retrieval and its integration with test-time adaptation in RAM-TTT.

**Test-Time Training.** Test-Time Training (TTT) adapts models to specific queries by updating parameters during inference (Sun et al., 2020; Xiao & Snoek, 2024). It has shown effectiveness in improving robustness across vision (Sun et al., 2020), vision-language (Shu et al., 2022; Yoon et al., 2024; Kojima et al., 2025), and sequence models (Gandelsman et al., 2022; Sun et al., 2024). Most recently, Test-Time Reinforcement Learning (TTRL) (Zuo et al., 2025) extends this paradigm to LLMs by framing test-time adaptation as an online reinforcement learning problem. While most TTT methods rely on self-supervised signals from test query, such approaches are not appropriate for complex multimodal reasoning in LVLMs under out-of-distribution scenarios. This gap motivates RAM-TTT, which unifies retrieval with robust test-time training to fully exploit retrieved examples while avoiding overfitting.

**Meta-Training.** Meta-learning aims to enable models to quickly adapt to new tasks with limited supervision by "learning to learn" (Finn et al., 2017). Recent advances extend this paradigm to language models: MetaICL (Min et al., 2021; Sinha et al., 2024) leverages meta-training across diverse tasks to improve few-shot In-Context Learning. While prior work applies meta-training mainly during pretraining or fine-tuning, our approach extends this perspective to test time, enabling retrieved examples to alternately serve as support sets and meta-queries for stronger adaptation under distribution shifts.

## 6 CONCLUSION

This paper introduces RAM-TTT, a retrieve–train–generate framework designed to enhance LVLMs on complex reasoning tasks at test time. By integrating LVLM Aligned Retrieval (LAR), which combines CLIP-based retrieval with LVLM confidence-based re-ranking, RAM-TTT selects examples that are both query-relevant and aligned with the LVLM's reasoning. Furthermore, RAM-TTT employs Meta Test-Time Training (Meta TTT), featuring a Contextual Align Strategy and ICL-style meta-updates, to adapt the LVLM with these examples while mitigating overfitting. Extensive experiments demonstrate consistent few-shot gains and robust generalization on complex multimodal reasoning.

ETHICS STATEMENT

This work focuses on developing methods to improve reasoning in Large Vision–Language Models (LVLMs) under distribution shifts through Retrieval-Augmented Meta Test-Time Training. Our experiments are conducted exclusively on publicly available benchmark such as MathVerse, LogicVista, and We-Math, which contain synthetic or publicly sourced problems without private or personally identifiable information. We do not use human subjects or sensitive data, and all dataset licenses and terms of use have been respected.

REPRODUCIBILITY STATEMENT

The paper provides full details of the RAM-TTT framework, including algorithmic design, training procedures, and evaluation protocols. Hyperparameter settings, prompt details, and dataset preprocessing steps are described in the Appendix. Ablation studies are included to validate design choices and robustness. All benchmarks (MathVerse, LogicVista, We-Math, MathVista) and datasets (MultiMath-300K, LLaVA-CoT-100k, Geo170K) used are publicly available and referenced with appropriate citations. To further support reproducibility, source code will be released soon.

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

## A  IMPLEMENTATION DETAILS

**Hardware and Software.**  Experiments were conducted on 4 NVIDIA A800 GPUs with 80GB memory each, using Ubuntu 20.04, Python 3.10, PyTorch 2.5.1+cu121, and ms-swift 3.1.0.dev0 (Zhao et al., 2024)[1].

**Training Hyperparameters.**  Unless otherwise specified, all experiments shared common training configurations. For model components, LLM parameters were unfrozen for training, while the ViT and Aligner components remained frozen.

The core training hyperparameters of our method are selected as follows:

- **Max sequence length:** 4096
- **LoRA rank:** 8
- **LoRA alpha:** 8
- **Batch size:** 1
- **Optimizer:** AdamW
- **Learning rate scheduler:** Cosine
- **Learning rate:** $5 \times 10^{-5}$ or $2 \times 10^{-5}$
- **Epochs:** 1 or 2
- **Mixed precision:** bf16
- **Random seed:** 42

The specific learning rates and number of training epochs are detailed as follows:

- **MathVerse:** learning rate $5 \times 10^{-5}$, 2 epochs
- **LogicVista:** learning rate $5 \times 10^{-5}$, 1 epoch
- **MathVista:** learning rate $2 \times 10^{-5}$, 2 epochs
- **We-Math:** learning rate $2 \times 10^{-5}$, 2 epochs

**Evaluation**  We evaluate all models using VLMEvalKit (Duan et al., 2024)[2], an open-source evaluation toolkit for large vision-language models, with Qwen-Turbo as the automatic judge instead of the default GPT-4o-mini.

## B  PROMPT DETAILS

**Zero-shot.**  For zero-shot inference, the model was provided with the image followed directly by the question, without any additional instructional text. Our prompt differs slightly from the standard VLMEvalKit setup, as we removed instructions such as: 'Please directly answer the question and provide the correct option letter, e.g., A, B, C, D.'

**In-Context Learning (ICL).**  The prompt format for ICL, which was consistent for both training (duringMeta TTT) and testing, is detailed in Figure 9.

**Confidence Score Calculation.**  When calculating confidence scores for the Model-Informed Re-ranking phase, the model was given the candidate support example and the test question. An additional instruction was provided to guide the model to output only the final answer. The specific prompt for this can be seen in Figure 10.

**Self-Correction.**  For the Self-Correction baseline, a specific instruction was designed to prompt the model to review its initial answer and make corrections if necessary. This instructional prompt is illustrated in Figure 11.

---

[1] https://github.com/modelscope/ms-swift
[2] https://github.com/open-compass/VLMEvalKit

You will be provided questions and answers similar to the ones below. Please study these examples carefully to understand the knowledge, concepts, and reasoning required to solve such problems. These examples serve as reference material containing important information. After the examples, I will give you a new question. Apply the knowledge you've learned from these examples to answer the new question accurately.

**Example 1**: <image> Question: {question 1} Answer: {answer 1}
**Example 2**: <image> Question: {question 2} Answer: {answer 2}
**Example 3**: <image> Question: {question 3} Answer: {answer 3}
**Example 4**: <image> Question: {question 4} Answer: {answer 4}

<image> Question: {test question}

Figure 9: Prompt format used for In-Context Learning (ICL) during training and testing.

You will be provided questions and answers similar to the ones below. Please study these examples carefully to understand the knowledge, concepts, and reasoning required to solve such problems. These examples serve as reference material containing important information. After the examples, I will give you a new question. Apply the knowledge you've learned from these examples to answer the new question accurately.

**Example 1**: <image> Question: {question 1} Answer: {answer 1}

<image> Question: {test question}

Please provide only your answer directly (e.g., a letter, a number, or a short phrase). Do not include any additional text or formatting.

Figure 10: Prompt format used for instructing the model to output only the final answer during confidence score calculation.

<image> Question: {test question}

My initial answer: {response}

Let me analyze my initial answer:
1. Is my answer accurate and complete?
2. Did I miss any important details in the image?
3. Are there any logical errors or inconsistencies in my reasoning?
4. Is my answer directly addressing what was asked?

I'll analyze step by step and then provide a refined answer:

Figure 11: Instructional prompt used for the Self-Correction baseline.

> You will be provided questions and answers similar to the ones below.
>
> **Example 1**: <image> Question: {question 1} Answer: {answer 1}
> **Example 2**: <image> Question: {question 2} Answer: {answer 2}
> **Example 3**: <image> Question: {question 3} Answer: {answer 3}
> **Example 4**: <image> Question: {question 4} Answer: {answer 4}
>
> <image> Question: {test question}

Figure 12: A simplified template that only keeps the minimal ICL structure.

> You will be provided questions and answers similar to the ones below. Let's think step by step to answer the last question accurately.
>
> **Example 1**: <image> Question: {question 1} Answer: {answer 1}
> **Example 2**: <image> Question: {question 2} Answer: {answer 2}
> **Example 3**: <image> Question: {question 3} Answer: {answer 3}
> **Example 4**: <image> Question: {question 4} Answer: {answer 4}
>
> <image> Question: {test question}

Figure 13: A "thinking" template that adds a standard "let's think step by step" instruction.

## C  RETRIEVAL CORPUS CONSTRUCTION

To build a large and diverse retrieval database for multimodal reasoning, we aggregate and process data from three major sources: MultiMath-300K (Peng et al., 2024), LLaVA-CoT-100k (Xu et al., 2024), and Geo170K (Gao et al., 2023). Below, we detail the filtering, deduplication, and standardization procedures applied to each dataset.

**MultiMath-300K.**  We use the full MultiMath-300K dataset, which contains a wide variety of multimodal math problems with detailed solutions. No additional filtering is applied, as the dataset already provides high-quality, well-structured question-answer pairs with accompanying images.

**LLaVA-CoT-100k.**  The LLaVA-CoT-100k dataset provides rich multimodal question-answer pairs, where each response is annotated with four pairs of special tags: `<SUMMARY>...</SUMMARY>`, `<CAPTION>...</CAPTION>`, `<REASONING>...</REASONING>`, and `<CONCLUSION>...</CONCLUSION>`. These tags serve distinct purposes:

- `<SUMMARY>`: Summarizes the overall approach or strategy for solving the problem.
- `<CAPTION>`: Describes relevant content or details from the associated image.
- `<REASONING>`: Contains the step-by-step reasoning or chain-of-thought process leading to the answer.
- `<CONCLUSION>`: Presents the final answer or conclusion derived from the reasoning.

For our retrieval corpus, we extract only the `<REASONING>` and `<CONCLUSION>` segments from each response. This choice is motivated by our focus on providing the LVLM with high-quality, explicit reasoning traces and clear final answers, which are most beneficial for in-context learning and test-time training. The `<SUMMARY>` and `<CAPTION>` tags, while informative, are omitted to maintain a concise and consistent format across all retrieval examples.

**Geo170K.**  Geo170K contains a mix of multimodal geometry problems, some with and some without explicit reasoning. We filter the dataset to retain only those entries annotated with chain-of-

thought (CoT) rationales. Additionally, to avoid redundancy and potential bias, we remove duplicate samples that share the same image, keeping only one instance per unique image.

**Data Standardization and Formatting.** After filtering, all samples from the three sources are reformatted into a unified structure to facilitate consistent in-context demonstration for the LVLM. Each example is organized as follows:

<think> *reasoning process* <\think> Answer: *final answer*

**Corpus Statistics.** The final retrieval corpus consists of 298,677 examples from MultiMath-300K, 98,571 from LLaVA-CoT-100k, and 5,962 from Geo170K, totaling 403,210 examples.

## D BENCHMARK DATASETS

We comprehensively evaluate our method on four diverse multimodal reasoning benchmarks, covering mathematical, logical, and spatial reasoning across varied contexts and task formats.

**MathVerse** (Zhang et al., 2024) A visual math benchmark emphasizing diagram understanding and reasoning under varying text–vision balances. We use the "testmini" split (3,940 samples), focusing on four subsets (TD, TL, VI, VD) with progressively less text and more visual reliance. The VO subset is excluded due to its image-only format.

**LogicVista** (Xiao et al., 2024) A logical reasoning benchmark with 448 questions spanning five tasks: Inductive, Deductive, Numerical, Spatial, and Mechanical reasoning. It supports both open-ended and multiple-choice evaluation, probing complementary reasoning skills in visual contexts.

**We-Math** (Qiao et al., 2024) A benchmark targeting principles of visual mathematical reasoning, with 1,740 samples. It introduces a four-dimensional metric—Insufficient Knowledge, Inadequate Generalization, Complete Mastery, and Rote Memorization—enabling fine-grained analysis beyond accuracy. We report results under the loose setting.

**MathVista** (Lu et al., 2023) A large-scale benchmark (1,000 "testmini" samples) covering algebra, geometry, statistics, logic, and scientific reasoning. It integrates 28 multimodal datasets and three new ones (IQTest, FunctionQA, PaperQA), spanning natural images, diagrams, charts, and synthetic scenes, thus ensuring broad coverage of visual-mathematical reasoning tasks.

## E THEORETICAL ANALYSIS

In this section, we provide theoretical underpinnings of the three core components of RAM-TTT: (1) the confidence score formulation, (2) the contextual alignment strategy, and (3) the meta test-time training paradigm. Together, these analyses establish a principled justification for our design choices.

### E.1 CONFIDENCE SCORE

Our confidence score is derived from the inverse perplexity of LVLM predictions. Perplexity is widely used to measure the uncertainty of language model outputs. Formally, given a target sequence $\hat{y} = (y_1, \ldots, y_T)$ and predictive distribution $p(y_t \mid x, y_{<t})$, the average negative log-likelihood (NLL) is

$$\mathcal{L} = -\frac{1}{T} \sum_{t=1}^{T} \log p(y_t \mid x, y_{<t}). \tag{7}$$

The perplexity is then defined as the exponential of this average NLL:

$$\text{PPL} = \exp(\mathcal{L}). \tag{8}$$

Intuitively, perplexity corresponds to the effective average branching factor of the model's predictive distribution: a lower perplexity indicates that the model assigns higher probability mass to the correct

sequence, i.e., lower uncertainty. This closely relates to the concept of Shannon entropy, as both quantify the uncertainty of a distribution, though perplexity is derived specifically from the model's conditional likelihoods over sequences.

We define the LVLM confidence score as the inverse perplexity:

$$\text{Conf} = \frac{1}{\text{PPL}}. \tag{9}$$

Thus, inverse perplexity provides a direct and training-free estimate of how compatible a candidate's reasoning trajectory is with the LVLM's predictive distribution, justifying its use as a proxy for reasoning utility during re-ranking.

### E.2 CONTEXTUAL ALIGNMENT

Our contextual alignment strategy (§2.3) can be understood through the lens of domain adaptation. Let $\mathcal{D}_{\text{train}}$ and $\mathcal{D}_{\text{test}}$ denote the training and inference distributions, respectively. Generalization bounds in domain adaptation theory show that the target error is bounded by the source error plus a divergence term between $\mathcal{D}_{\text{train}}$ and $\mathcal{D}_{\text{test}}$. By augmenting the support set with $(x_q, \tilde{y}_q)$, we explicitly reduce the divergence between training and inference contexts. This alignment ensures that optimization at test time is consistent with the eventual inference setting, lowering the risk of distribution mismatch.

### E.3 META TEST-TIME TRAINING

Meta TTT is theoretically grounded in meta-learning frameworks such as MAML (Finn et al., 2017). In MAML, tasks are structured into support and query sets, and the objective is to minimize the expected risk across tasks:

$$\min_{\theta} \; \mathbb{E}_{\mathcal{T} \sim p(\mathcal{T})} \left[ \mathcal{L}_{\mathcal{T}_{\text{query}}}(U(\theta, \mathcal{T}_{\text{support}})) \right], \tag{10}$$

where $U$ denotes the inner update. In our formulation, retrieved examples alternately serve as support and meta-query sets, mimicking this meta-learning structure at test time. The leave-one-out construction of $\mathcal{D}_{\text{meta}}$ ensures that adaptation minimizes expected error across all retrieved examples, thereby regularizing against overfitting to any single support instance.

## F COMPUTATIONAL OVERHEAD ANALYSIS

We provide a detailed analysis of the per-query computational cost of each component in RAM-TTT. All measurements were conducted on a single NVIDIA A800 GPU using Qwen2-VL-7B as the backbone.

Table 6 summarizes the average latency per query for each stage of our framework. Meta TTT constitutes the dominant cost due to parameter updates performed for each test query, while retrieval and re-ranking account for a comparatively small portion of total latency.

Table 6: Per-query latency of each component in RAM-TTT.

| Component | Avg. Latency (s) |
|---|---|
| CLIP-based retrieval | 0.43 |
| LVLM re-ranking | 4.31 |
| Contextual alignment | 0.59 |
| Meta TTT | 16.17 |
| Final inference | 4.42 |
| **Total (RAM-TTT)** | **25.92** |

Although RAM-TTT introduces additional computation compared to zero-shot and ICL baselines, Figure 6 demonstrates that the method achieves a substantially better accuracy-compute frontier than other test-time scaling strategies. This indicates that the additional computation is efficiently converted into improved reasoning performance.

## G  CASE STUDY

**Question:**
Find the surface area of the rectangular prism with length
equal to 12 m.

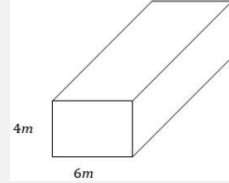

**Zero-shot(×):**
The surface area of a rectangular prism is 2 times the product of the length and the width plus 2 times the product of the length and the height plus 2 times the product of the width and the height. In this case, the length is 12 m and the width is 4 m. We don't know the height, but we can find it by using the fact that the volume of a rectangular prism is equal to the length times the width times the height. In this case, the volume is equal to 12 m times 4 times the height, or 48 m^3. We can then solve for the height by dividing both sides of the equation by 12 m times 4 m, which gives us the height of 2 m. Now we can use the height to find the surface area. The surface area is 2 times the product of the length and the width plus 2 times the product of the length and the height plus 2 times the product of the width and the height, or 2 times 12 m times 4 m plus 2 times 12 m times 2 m plus 2 times 4 m times 2 m, which is equal to 160 m^2.

**ICL(×):**
<think>Step 1 (Surface area formula): According to the surface area formula of a rectangular prism, the surface area is the sum of the areas of each face.
Step 2 (First two faces area): Each 4×4 square has an area of 4*4 = 16, and there are two of these squares, so we have 2*16.
Step 3 (Bottom and top face areas): The length is 10, the width is 4, so each bottom and top face has an area of 10*4 = 40, and the two bottom and top faces have a total area of 2*40 = 80.
Step 4 (Side face areas): The length is 12, the width is 4, so each side face has an area of 12*4 = 48, and the two side faces have a total area of 2*48 = 96.
Step 5 (Total surface area): The total surface area is the sum of the areas of each face, so the total surface area is 2*16 + 80 + 96 = 208.
</think>Answer: \boxed{208}

**RAM-TTT(✓):**
<think>Step 1 (Surface area formula): The formula for the surface area of a rectangular prism is 2(Length × Width + Length × Height + Width × Height).
Step 2 (Substitute the values): 2(12 m × 4 m + 12 m × 6 m + 4 m × 6 m).
Step 3 (Calculate individual areas): 2(48 m² + 72 m² + 24 m²).
Step 4 (Calculate surface area): 2(144 m²) = 288 m².</think>
Answer: \boxed{288 m²}

Figure 14: Case study 1 of Zero-shot, ICL, and RAM-TTT for Reasoning

As shown in Figure 14, this comparison reveals a clear progression in reasoning quality. The Zero-shot's reasoning is chaotic; it correctly states the formula but fails to parse the inputs, leading to hallucination and a nonsensical result. ICL improves by learning the step-by-step format, but demonstrates poor context utilization by incorrectly mixing data from the examples into the current query's calculation. In contrast, RAM-TTT delivers a concise, logical, and correct solution. By adapting to the retrieved examples, our method enables the model to not just mimic the format but to properly learn and apply the reasoning pattern to the specific data of the query at hand, clearly demonstrating an enhanced ability to generate deeper and more accurate reasoning steps.

**Question:**

Question: As shown in the figure, it is known that angle A = 80.0, angle B = 60.0, DE parallel BC, then the size of angle CED is ()

Choices:

A:40°

B:60°

C:120°

D:140°

**Zero-shot(✗):**

A

**ICL(✗):**

<think>Step 1 (Triangle Exterior Angle): Calculate the exterior angle of ∠A: ∠A + ∠ACB = 180°.

Step 2 (Substitute the known angles): 80° + ∠ACB = 180°.

Step 3 (Calculate the difference): ∠ACB = 180° - 80° = 100°.

Step 4 (Parallel lines property): Since DE ∥ BC, based on the property of alternate interior angles being equal, ∠CED = ∠ACB = 100°.

</think>Answer: \boxed{100°}

**RAM-TTT(✓):**

<think>Step 1 (Sum of angles theorem): In △ABC, ∠A + ∠B + ∠C = 180°.

Step 2 (Substitute known angles): ∠A = 80°, ∠B = 60°.

Step 3 (Calculate ∠C): ∠C = 180° - 80° - 60° = 40°.

Step 4 (Alternate interior angles with parallel lines): Since DE∥BC, ∠CED and ∠C are alternate interior angles and therefore equal.

Step 5 (Determine ∠CED value): ∠CED = 180° - ∠C = 180° - 40° = 140°.

</think>Answer: \boxed{140°}

Figure 15: Case study 2 of Zero-shot, ICL, and RAM-TTT for Reasoning

Similarly, Figure 15 further illustrates the robustness of RAM-TTT across different problem settings. In this case, Zero-shot once again produces a superficial and incorrect answer, lacking any structured reasoning. ICL shows partial improvement by attempting a structured derivation but still introduces errors. By contrast, RAM-TTT integrates retrieval with task-specific adaptation, producing step-by-step reasoning that is both coherent and mathematically accurate. This highlights that RAM-TTT does not merely imitate prior demonstrations but actively transfers and contextualizes reasoning strategies, ensuring reliability across diverse reasoning challenges.

## H  THE USE OF LARGE LANGUAGE MODELS (LLMs)

We used LLMs as an assistive tool during this work. Specifically, LLMs were employed to help refine parts of the manuscript, and to provide limited coding assistance during experiments. All research ideas, analyses, and final decisions were made by the authors, who take full responsibility for the content.

