# OpenReview forum: "Retrieval-Augmented Meta Test-Time Training for Multimodal Reasoning"
_ICLR.cc/2026/Conference — Submitted to ICLR 2026_

### Official Review · Reviewer_PSrm · 2025-10-21

**Soundness:** 2
**Presentation:** 3
**Contribution:** 2
**Rating:** 4
**Confidence:** 4

**Summary:**

This paper presents a multimodal retrieval-augmented generation framework, which consists of two components: LVLM Aligned Retrieval (LAR) and Meta Test-Time Training (Meta TTT). Overall, the paper is well presented. However, several technical shortcomings remain.

**Strengths:**

1. The paper is well presented and easy to follow.
2. Additional experiment on Phi-3.5-Vision and Pixtral-12B demonstrate the method's generalization across different model architectures.

**Weaknesses:**

1. The paper proposes LAR and Meta TTT, while the title emphasizes only Meta TTT. If Meta TTT is the core contribution, I recommend focusing solely on it and removing LAR to avoid diluting the main contribution.
2. While both LAR and Meta TTT are shown to be effective, the paper lacks a deeper analysis of how they work. For example, the authors claim that LAR improves reasoning utility, but no ablation or analysis is provided to support this. The same applies to Meta TTT.
3. The work emphasizes multimodal reasoning, but the backbone model (Qwen2-VL) does not support CoT reasoning, which raises concerns about how reasoning is truly being evaluated. Moreover, given the recent release of Qwen2.5/3-VL, experiments on these stronger models would be more compelling.
4. There is almost no comparison with other state-of-the-art methods in multimodal RAG. The comparisons with ICL and TTT in Tables 1 and 2 can be seen as ablation study.
5. I also question the suitability of the selected primary areas "transfer learning, meta-learning, and lifelong learning" for this work, as the paper is fundamentally centered on multimodal RAG.

**Questions:**

See Weakness.

---

> ### Author Response · Authors · 2025-11-22
> **Response to Reviewer PSrm (Part 1)**
>
> We sincerely thank Reviewer PSrm for the constructive feedback and positive remarks regarding the clarity of presentation and the cross‑backbone experiments. Below we address each concern in turn.
>
> **W1: The paper proposes LAR and Meta TTT, while the title emphasizes only Meta TTT. If Meta TTT is the core contribution, I recommend focusing solely on it and removing LAR to avoid diluting the main contribution.**
>
> **A:** We appreciate this comment and agree that our exposition may give the impression of two separate contributions. Our intention is that RAM‑TTT is a unified retrieve–train–generate framework, in which LAR is the retrieval stage and Meta TTT is the adaptation stage.
>
> Empirically, Meta TTT alone, when applied to CLIP‑retrieved examples, already improves over standard TTT (Table 3, 32.9→35.8 on MathVerse TL), and plugging in LAR further boosts performance to 38.5 (+5.6 over TTT). This indicates that Meta TTT depends on having retrieval that is aligned with the LVLM's reasoning, which is exactly what LAR is designed to provide.
>
> To avoid confusion, we will (i) make this dependency explicit in Sec. 1 and Sec. 2 by describing LAR as "the retrieval stage of RAM‑TTT" rather than as a separate method, and (ii) compress low‑level details of LAR into the appendix so that Meta TTT and the overall RAM‑TTT framework remain the main narrative focus.
>
> **W2: While both LAR and Meta TTT are shown to be effective, the paper lacks a deeper analysis of how they work. For example, the authors claim that LAR improves reasoning utility, but no ablation or analysis is provided to support this. The same applies to Meta TTT.**
>
> **A:** Thank you for pointing this out. We agree that the current analysis can be made more explicit.
>
> For LAR, we clarify in the revised version that its contribution goes beyond semantic relevance by explicitly connecting our claims to existing results:
> - Table 3 shows that, on top of Meta TTT, adding confidence‑based re‑ranking brings a further gain.
> - Figures 3 and 5 show that LAR+ICL consistently outperforms CLIP‑Retrieval+ICL across retrieval ratios r and support set sizes k. They also show that RAM‑TTT continues to benefit from additional examples while baselines plateau or degrade, indicating that the selected examples are genuinely useful for reasoning rather than only superficially similar.
>
> We will provide a more in-depth analysis of these phenomena in the ablation section and discuss why LAR works in Appendix E.
>
> For Meta TTT, we will (i) strengthen Sec. 3 by providing a deeper analysis of the improvements over standard TTT shown in Table 3 and Figure 5; (ii) add a new subsection in Sec. 4 ("Adaptation Stability and Overfitting Analysis") with the corresponding curves (Figure 7), showing that standard TTT quickly overfits, weight decay only mildly helps, while Meta TTT remains stable across a wide range of adaptation steps which directly supports its robustness to avoid overfitting; and (iii) discuss why Meta TTT works in Appendix E.
>
> **W3: The work emphasizes multimodal reasoning, but the backbone model (Qwen2-VL) does not support CoT reasoning, which raises concerns about how reasoning is truly being evaluated. Moreover, given the recent release of Qwen2.5/3-VL, experiments on these stronger models would be more compelling.**
>
> **A:** We thank the reviewer for raising this concern.
>
> First, Qwen2‑VL is capable of step‑by‑step reasoning when prompted. In our setting, all retrieved examples are formatted as "<think> reasoning process </think> Answer: final answer", which elicits chain‑of‑thought style outputs from the backbone. Appendix G already shows qualitative examples where the model produces explicit reasoning steps and RAM‑TTT improves both the intermediate reasoning and the final answers over zero‑shot and ICL.
>
> Second, we agree that testing stronger backbones is important. Our current experiments already demonstrate that RAM‑TTT yields consistent gains on two architecturally different LVLMs, Phi‑3.5‑Vision and Pixtral‑12B (Fig. 4), supporting the backbone‑agnostic nature of the framework. We further include experiments on Qwen2.5-VL-3B, Qwen2.5-VL-7B and Qwen3-VL-4B. As summarized in the table below, RAM‑TTT consistently improves over both zero-shot and ICL. These results further confirm that RAM‑TTT scales well with backbone strength and remains effective when moving to newer Qwen2.5/3‑VL models.
>
> | Model| Method| MathVerse (TL) | MathVista (GPS) |
> |----------------|-------|-------|------|
> | Qwen2.5-VL-3B  | Zero-shot  | 32.1           | 51.4 |
> | | ICL        | 34.3           | 55.3             |
> | | RAM-TTT    | **37.8**       | **59.1**         |
> | Qwen2.5-VL-7B  | Zero-shot  | 42.6           | 64               |
> | | ICL        | 44.3           | 69.7             |
> | | RAM-TTT    | **45**         | **72.1**         |
> | Qwen3-VL-4B    | Zero-shot  | 49         | 75.5             |
> | | ICL| 51.1           | 73.1             |
> | | RAM-TTT    | **54**         | **76.4**       |

---

> > ### Author Response · Authors · 2025-11-22
> > **Response to Reviewer PSrm (Part 2)**
> >
> > **W4: There is almost no comparison with other state-of-the-art methods in multimodal RAG. The comparisons with ICL and TTT in Tables 1 and 2 can be seen as ablation study.**
> >
> > **A:** Thank you for this comment. Our current baselines labeled "ICL” correspond to the standard multimodal RAG pipeline: CLIP‑based retrieval followed by in‑context generation without parameter updates.
> >
> > Conceptually, RAM‑TTT is complementary to existing multimodal RAG methods rather than a competitor: any stronger multimodal retriever can replace our Stage‑1 CLIP retriever to produce the candidate set, while LAR and Meta TTT remain unchanged and still bring gains. To make this complementarity more concrete, Table 4(a) already demonstrates that plugging in a stronger CLIP backbone as the retriever (e.g., upgrading from ViT‑B/32 to ViT‑L/14@336px) yields consistent gains for RAM‑TTT on MathVerse, indicating that our framework can directly benefit from advances in multimodal retrieval.
> >
> > **W5: I also question the suitability of the selected primary areas "transfer learning, meta-learning, and lifelong learning" for this work, as the paper is fundamentally centered on multimodal RAG.**
> >
> > **A:** We understand the reviewer's concern. While our framework integrates retrieval, the key methodological novelty is on the adaptation side rather than on retrieval itself.
> >
> > TTT is fundamentally a transfer learning paradigm: for each query, it retrieves a small support set and then updates the LVLM parameters at inference using a supervised loss over this support set, adapting the pretrained model to the test‑time distribution defined by the benchmark and its retrieved context. Meta TTT further introduces a meta‑learning objective, where retrieved examples alternately act as support and meta‑query sets in a leave‑one‑out episodic construction of $D_{\text{meta}}$, in the spirit of model‑agnostic meta‑learning (MAML), which empirically improves robustness and sample efficiency.
> >
> > By contrast, the retrieval module (LAR) is intentionally simple and replaceable: it uses CLIP for Stage‑1 retrieval and a lightweight LVLM confidence‑based re‑ranking in Stage‑2, and Table 4(a) shows that stronger retriver backbones can be plugged in to further boost RAM‑TTT without changing the core algorithm.
> >
> > For these reasons, categorizing our work under "transfer learning” and "meta-learning" more accurately reflects its primary methodological contributions.

---

> ### Comment · Reviewer_PSrm · 2025-11-24
> **Response to the rebuttal**
>
> Thank you for the rebuttal. However, I still believe that the contribution (W1), reasoning (W2) and the experiments (W4)  need improvement in the submitted manuscript. Therefore, I will maintain my score.

---

### Official Review · Reviewer_XVo3 · 2025-10-29

**Soundness:** 2
**Presentation:** 2
**Contribution:** 3
**Rating:** 4
**Confidence:** 4

**Summary:**

This paper introduces RAM-TTT, a Retrieval-Augmented Meta Test-Time Training framework aimed at enhancing the reasoning capability of Large Vision-Language Models (LVLMs) under distribution shifts. RAM-TTT unifies two key innovations: (1) LVLM-Aligned Retrieval (LAR) selects support examples based on both semantic relevance and their utility for reasoning, and (2) Meta Test-Time Training (Meta TTT) leverages a meta-learning paradigm to adapt the LVLM with retrieved examples while mitigating overfitting.

**Strengths:**

Motivational Clarity and Problem Significance. The paper identifies and articulates a genuine gap: standard Test-Time Scaling methods in LVLMs typically fail to incorporate new, external knowledge, which limits adaptation to distribution shifts and complex reasoning scenarios. Targeting this problem is valuable for the field.

Extensive Empirical Validation. Experiments span four strong reasoning benchmarks and three LVLM backbones. Results in Table 1 and Table 2 show RAM-TTT outperforming several reasonable baselines (e.g., Zero-shot, Self-Correction, TTT, ICL) by a notable margin.

Detailed Ablations and Analyses. Ablation results are comprehensive, examining the importance of each module (Meta TTT, LAR, contextual alignment), retriever variants, permutation effects, and computational trade-offs, demonstrating careful empirical study. Figures 3 and 4, for instance, solidly illustrate the method’s robustness to parameter settings and model backbone.

**Weaknesses:**

1. In Figure 2, essential mathematical annotations are missing. For example, the (e_i) referenced in §2.2 and the candidate set (C_{\text{cand}}) do not appear in the figure.
2. Line 173 uses the inverse of perplexity to measure the LVLM’s confidence, but this lacks a detailed explanation (although I found one in the supplementary material) and omits citations to key references:
   1. *Language Models are Unsupervised Multitask Learners* (GPT-2)
   2. *Calibrating Trust of Large Language Models via Perturbation-Based Calibration*
   3. *Self-Consistency Improves Chain of Thought Reasoning in Language Models*
3. The treatment of overfitting remains insufficient. Although the Meta TTT paradigm is designed to avoid overfitting on tiny support sets, the evidence for robustness is largely anecdotal (see the discussion around Figure 5). The scientific value would be improved by providing more rigorous quantification—e.g., plotting performance/error versus the number of adaptation steps, or comparing against alternative regularization schemes.
4. The ablation tables (especially Tables 3–5) do not report standard deviations or confidence intervals across multiple runs.
5. RAM-TTT introduces additional computation for re-ranking, meta-training, and prompt permutations. However, the paper lacks a detailed discussion of the framework’s complexity and computational overhead.
6. I notice the authors use proprietary prompt designs. It remains unclear how the method performs under unseen or alternative prompts.

**Questions:**

The same as Weaknesses.

---

> ### Author Response · Authors · 2025-11-22
> **Response to Reviewer XVo3 (Part 1)**
>
> We sincerely thank Reviewer XVo3 for the careful evaluation and insightful comments. We are encouraged that the reviewer found the problem motivation meaningful and appreciated the breadth of empirical validation and ablation studies. The feedback on methodological clarity and analysis depth is highly valuable. We respond point-by-point to all identified weaknesses and questions below.
>
> **W1: In Figure 2, essential mathematical annotations are missing. For example, the $e_i$ referenced in §2.2 and the candidate set $C_{\text{cand}}$ do not appear in the figure.**
>
> **A:** Thank you for catching this inconsistency. In the original manuscript, the notation $e_i$ in Sec. 2.2 was a typo and was redundant. The blue blocks $(x_1,y_1),(x_2,y_2),...,(x_n,y_n)$ in Figure 2 correspond to the candidate set $C_{\text{cand}}$. In the revision, we will (i) removed $e_i$ from the text and (ii) explicitly label the blue blocks as the candidate set $C_{\text{cand}}$.
>
> This makes the notation in Figure 2 fully consistent with Section 2.2.
>
> **W2: Line 173 uses the inverse of perplexity to measure the LVLM's confidence, but this lacks a detailed explanation (although I found one in the supplementary material) and omits citations to key references.**
>
> **A:** We appreciate the reviewer's suggestion to better justify and reference our confidence measure.
> Conceptually, lower perplexity means that the LVLM assigns higher probability mass to its own generated sequence, i.e., it is more certain about this reasoning trajectory. Using $1/PPL$ as a score therefore provides a training‑free estimate of how well a candidate example aligns with the model's internal predictive distribution. This interpretation is consistent with recent work that uses model generation probabilities as a proxy for self-estimated confidence[1,2]
>
> Empirically, confidence-based re‑ranking consistently improves performance across retrieval ratios and backbones (Figure 3 and Figure 5), indicating that this signal is stable and useful for selecting reasoning‑helpful examples.
>
> In the revision, we will make this connection explicit in Section 2.2 and briefly point to the more detailed derivation in Appendix E. We will also add the missing references suggested by the reviewer(GPT2 and Self-Consistency).
>
> [1] A Survey of Confidence Estimation and Calibration in Large Language Models (Geng et al., NAACL 2024)
>
> [2] Active Retrieval Augmented Generation (Jiang et al., EMNLP 2023)
>
>
> **W3: The treatment of overfitting remains insufficient. Although the Meta TTT paradigm is designed to avoid overfitting on tiny support sets, the evidence for robustness is largely anecdotal (see the discussion around Figure 5). The scientific value would be improved by providing more rigorous quantification—e.g., plotting performance/error versus the number of adaptation steps, or comparing against alternative regularization schemes.**
>
> **A:** Thank you for pointing out the need for a more rigorous treatment of overfitting. We have conducted additional experiments that explicitly track performance as a function of the number of adaptation steps.
> Concretely, we now compare three variants: (i) TTT, (ii) TTT with weight decay, and (iii) RAM‑TTT.
> On MathVista GPS, TTT improves in the first few steps but then quickly overfits: accuracy peaks at 44.7% around step 8 and drops to 41.3% by step 16. TTT with weight decay delays this collapse but still degrades after about 32 steps. In contrast, RAM‑TTT improves steadily in the early steps and remains stable: its accuracy stays within ±0.5% of the peak even after 16 adaptation steps.
>
> In the revised manuscript, we will add a subsection "Adaptation Stability and Overfitting Analysis” in Section 4 and include a new Figure (Fig. 7) showing these performance–versus–steps curves. This directly supports our claim that the Meta TTT paradigm mitigates overfitting compared to standard TTT and simple regularization.

---

> > ### Author Response · Authors · 2025-11-22
> > **Response to Reviewer XVo3 (Part 2)**
> >
> > **W4: The ablation tables (especially Tables 3–5) do not report standard deviations or confidence intervals across multiple runs.**
> >
> > **A:** Thank you for pointing this out. We will add results with standard deviations in the appendix. Below we include representative results, and our conclusions remain unchanged.
> >
> > | Method   | MathVerse    | LogicVista      |
> > |----------|---------------------|------------------|
> > | TTT        |        33.7 ± 0.2         |    31.1 ± 0.1    |
> > | + Meta TTT |        36.6 ± 0.6      |     34.5 ± 0.5      |
> > | + Conf. Re-ranking |   38.0 ± 0.4   |       35.3 ± 0.2    |
> > | + Contextual Align |    39.6 ± 0.7  |       35.8 ± 0.2    |
> >
> > **W5: RAM-TTT introduces additional computation for re-ranking, meta-training, and prompt permutations. However, the paper lacks a detailed discussion of the framework's complexity and computational overhead.**
> >
> > **A:** We agree that RAM‑TTT introduces additional computation, and we now provide a more detailed breakdown. On Qwen2‑VL‑7B with a single NVIDIA A800 GPU, the average per‑query latency is:
> >
> > | Component            | Avg. Latency (s) |
> > | -------------------- | ---------------- |
> > | CLIP-based retrieval | 0.43             |
> > | LVLM re-ranking      | 4.31             |
> > | Contextual alignment | 0.59             |
> > | Meta TTT             | 16.17            |
> > | Final inference      | 4.42             |
> > | **Total (RAM-TTT)**  | **25.92**        |
> >
> > The dominant cost comes from Meta TTT, which performs query‑specific parameter updates. The other components are relatively lightweight. As shown in Figure 6, despite this additional cost, RAM‑TTT lies on a significantly better accuracy–time frontier than Zero‑shot, ICL, Self‑Correction, and standard TTT: for similar or slightly higher compute, it delivers the largest accuracy gains. We will include this breakdown and discussion in an appendix.
> >
> > **W6: I notice the authors use proprietary prompt designs. It remains unclear how the method performs under unseen or alternative prompts.**
> >
> > **A:** We appreciate the concern about prompt dependence. Our method does not require any proprietary or heavily engineered prompts: it only assumes that the same template is used for Meta TTT and final inference.
> > To verify robustness, we evaluated RAM‑TTT with three prompt templates: (i) the original template used in the paper, (ii) a simplified template that only keeps the minimal ICL structure, and (iii) a "thinking” template that adds a standard "let's think step by step” instruction.
> >
> > Original Prompt
> > ```
> > You will be provided questions and answers similar to the ones below.
> > Please study these examples carefully to understand the knowledge, concepts,
> > and reasoning required to solve such problems. After the examples, I will give
> > you a new question. Apply the knowledge you have learned from these examples
> > to answer the new question accurately.
> >
> > Example 1: <image> Question: {question 1} Answer: {answer 1}
> > Example 2: <image> Question: {question 2} Answer: {answer 2}
> > Example 3: <image> Question: {question 3} Answer: {answer 3}
> > Example 4: <image> Question: {question 4} Answer: {answer 4}
> > <image> Question: {test question}
> > ```
> > Simple Prompt
> > ```
> > You will be provided questions and answers similar to the ones below.
> >
> > Example 1: <image> Question: {question 1} Answer: {answer 1}
> > Example 2: <image> Question: {question 2} Answer: {answer 2}
> > Example 3: <image> Question: {question 3} Answer: {answer 3}
> > Example 4: <image> Question: {question 4} Answer: {answer 4}
> > <image> Question: {test question}
> > ```
> > Thinking Prompt
> > ```
> > You will be provided questions and answers similar to the ones below.
> > Let's think step by step to answer the last question accurately.
> > Example 1: <image> Question: {question 1} Answer: {answer 1}
> > Example 2: <image> Question: {question 2} Answer: {answer 2}
> > Example 3: <image> Question: {question 3} Answer: {answer 3}
> > Example 4: <image> Question: {question 4} Answer: {answer 4}
> > <image> Question: {test question}
> > ```
> >
> > We then evaluated RAM-TTT under these templates. The results are:
> >
> > | Prompt           | MathVista (GPS) | LogicVista   |
> > |------------------|-----------------|--------------|
> > | Original Prompt  | 52.4 ± 1.0      | 35.8 ± 0.2   |
> > | Simple Prompt    | 50.7 ± 0.6      | 35.0 ± 0.9   |
> > | Thinking Prompt  | 53.0 ± 0.8      | 37.5 ± 0.5   |
> >
> > These results show that RAM-TTT maintains stable and strong performance across different prompt styles. The variation across prompts is small, demonstrating that the method is not sensitive to prompt wording.

---

> > > ### Comment · Reviewer_XVo3 · 2025-11-27
> > > **Thank you for the detailed rebuttal. The authors have thoroughly addressed my previous comments, and I have no further questions. That said, I still find the core contribution of the paper somewhat limited. Although the results are strong, they come at a relatively high cost. I will therefore maintain my original score.**
> > >
> > > Thank you for the detailed rebuttal. The authors have thoroughly addressed my previous comments, and I have no further questions. That said, I still find the core contribution of the paper somewhat limited. Although the results are strong, they come at a relatively high cost. I will therefore maintain my original score.
> > > 1. From the authors’ response in W5, it is clear that the proposed model incurs a relatively high cost.
> > > 2. In addition, the paper’s main contributions are not clearly articulated and read more like a combination of several existing techniques rather than a well-defined, unified contribution.

---

### Official Review · Reviewer_6SwX · 2025-10-31

**Soundness:** 3
**Presentation:** 2
**Contribution:** 3
**Rating:** 6
**Confidence:** 3

**Summary:**

This paper proposes a novel Retrieval-Augmented Test-Time Training (RAM-TTT) framework that successfully integrates the benefits of Retrieval-Augmented Generation (RAG) with Test-Time Training (TTT) techniques. The framework’s design moves beyond simple semantic similarity by additionally incorporating "reasoning utility" into the training process, thereby ensuring retrieved results are optimal for complex reasoning tasks. The experiments demonstrate that this new framework effectively enhances answer accuracy. However, the paper lacks sufficient clarification regarding its core motivation, and the claims about the effectiveness of the reasoning utility metric require more substantial evidence.

**Strengths:**

1）Novel Integration: The paper introduces a novel framework (RAM-TTT) that effectively integrates Retrieval-Augmented Generation (RAG) with Test-Time Training (TTT), which is a promising and timely direction for enhancing model performance, particularly under distribution shifts.

2）Reasoning-Aware Retrieval: The training framework is uniquely designed to optimize retrieval not only based on traditional semantic similarity but also on a new metric, "reasoning utility," ensuring the retrieved examples are highly relevant to the complex reasoning task at hand.

3）Empirical Performance: The experimental results demonstrate that the proposed method significantly improves the accuracy of answers across various benchmarks, validating the effectiveness of the combined approach.

**Weaknesses:**

1）The core motivation and justification for the proposed method could be elaborated more thoroughly.

2）While the abstract claims improvements in both accuracy and reasoning efficiency, the experimental section heavily prioritizes verification of accuracy, with insufficient evidence or analysis dedicated to demonstrating the gains in reasoning efficiency.

3）The analysis and interpretation of the observed phenomena and hypotheses within the ablation study are too brief and could benefit from deeper discussion.

**Questions:**

1）Why do you choose high-confidence pairs in stage 1? These pairs are used for training. If the trained data is low-confident which means the model haven't learned them, will the performance be better? Because the reference in RAG is often supplementary material which the model haven't seen, if model have studied it in advance, the performance will be better in most cases.

2）In stage 2, why do you use the leave-one-out strategy to make $M$ random permutations? For data enhancement?

3）Can you briefly describe the difference between TTT and meta TTT?

---

> ### Author Response · Authors · 2025-11-22
> **Response to Reviewer 6SwX (Part 1)**
>
> We sincerely thank Reviewer 6SwX for the thoughtful and constructive review. We appreciate the reviewer's recognition that our method is both novel and effective. The detailed feedback is truly encouraging. Below, we address each of the raised weaknesses and questions in turn.
>
> **W1: The core motivation and justification for the proposed method could be elaborated more thoroughly.**
>
> **A:** We thank the reviewer for highlighting the need for clearer motivation. Our core motivation is that while Test-Time Scaling improves reasoning, it still operates entirely on internal model knowledge, which makes it fragile under distribution shifts where the model lacks the necessary reasoning patterns or domain knowledge. RAM-TTT is designed to address this by (i) enabling test-time scaling to acquire external reasoning evidence and (ii) providing a learning paradigm that can internalize such evidence in a stable and generalizable manner.
>
> Concretely, LAR goes beyond semantic similarity and uses the LVLM confidence–based reasoning utility to select examples that not only semanticly similar to the query but also align well with the model's reasoning. Meta TTT then treats these examples as alternating support and meta-query sets at test time, which mitigates overfitting to a tiny support set and lets the model "learn how to learn” from retrieved reasoning traces.
>
> In the revised version, We will make this motivation more explicit by (i) refining the sentence following the discussion of TTS limitations to directly state our goal of enabling test-time scaling to incorporate external reasoning signals, and (ii) slightly revising the opening of the retrieval paragraph to clarify how retrieval partially alleviates this limitation and motivates the design of RAM-TTT.
>
> **W2: While the abstract claims improvements in both accuracy and reasoning efficiency, the experimental section heavily prioritizes verification of accuracy, with insufficient evidence or analysis dedicated to demonstrating the gains in reasoning efficiency.**
>
> **A:** We thank the reviewer for this comment. In our work, "reasoning efficiency” refers to how effectively extra test-time compute and retrieved examples are converted into reasoning accuracy gains. Section 4 analyzes this in two ways. First, Figure 5 varies the support set size k and shows that as k increases from 2 to 6, RAM-TTT continues to improve, whereas ICL and LAR+ICL plateau or even degrade. This indicates that RAM-TTT makes more efficient use of additional retrieved examples, turning them into useful reasoning signals rather than noise.
>
> Second, Figure 6 reports the accuracy–time trade-off across Zero-shot, ICL, Self-Correction, TTT, and RAM-TTT. At comparable average test-time per query, RAM-TTT achieves higher accuracy than TTT and Self-Correction, which consume substantial compute but offer limited gains. In other words, RAM-TTT consistently lies on a more favorable accuracy–compute frontier, showing that additional FLOPs at test time are effectively translated into genuine reasoning improvements.
>
> In addition, compared to standard supervised fine-tuning that require substantial domain-specific training data, RAM-TTT adapts entirely at test time using only a small retrieved support set. This enables the model to acquire domain-relevant reasoning signals without any heavy offline training.
>
> **W3: The analysis and interpretation of the observed phenomena and hypotheses within the ablation study are too brief and could benefit from deeper discussion.**
>
> **A:** In the revised manuscript, we will expand Sec. 3.3 by adding interpretation after each ablation result. For example, after Table 3 we will explain that Meta TTT reduces overfitting because its episodic structure forces updates to generalize across retrieved examples rather than memorizing any single instance, substantially reducing single-example overfitting and improving adaptation stability. Similarly, for the retrieval ratio r and permutation times M (Tables 4–5), we will explicitly point out that overly large candidate pools or too many permutations introduce noisy or redundant episodes, which explains the non-monotonic trends.
>
> In addition, we will add a new subsection "Adaptation Stability and Overfitting Analysis” in Sec. 4, including a performance-vs-adaptation-steps plot comparing TTT, TTT with weight decay, and RAM-TTT. This plot shows that TTT rapidly overfits after a few steps, weight decay only slightly delays this degradation, while RAM-TTT improves in the early steps and then remains stable, providing a more quantitative explanation of why the meta-learning objective yields robust test-time adaptation.

---

> ### Author Response · Authors · 2025-11-22
> **Response to Reviewer 6SwX (Part 2)**
>
> **Q1: Why do you choose high-confidence pairs in stage 1? These pairs are used for training. If the trained data is low-confident which means the model haven't learned them, will the performance be better? Because the reference in RAG is often supplementary material which the model haven't seen, if model have studied it in advance, the performance will be better in most cases.**
>
> **A:** We appreciate the reviewer's question.
> We choose high-confidence pairs because, in our experiments, low-confidence examples behave like noisy or even adversarial supervision: training on them significantly hurts test-time adaptation, sometimes even below the zero-shot baseline.
>
> Importantly, our confidence score is not measuring whether the model has "already learned” the example. As defined in Sec. 2.2 and Appendix E, for each candidate $(x_i, y_i)$, we compute an LVLM confidence score:
> $$\text{Conf}_q^{(i)}=\frac{1}{\text{PPL}_q^{(i)}}$$
>
> on the query-conditional prompt:$[I_{\text{dir}}, x_i, y_i, x_q]$
>
> A higher score means that the candidate's reasoning pattern leads to low predictive uncertainty on the query, i.e., it provides a reasoning trajectory that the LVLM can reliably follow for the specific query.
>
> To directly test the reviewer's hypothesis, we ran an additional experiment where we invert the confidence selection and deliberately use the lowest-confidence candidates for adaptation (6-shot setting). The results are:
>
> | Method                               | MathVista (GPS) | LogicVista |
> | ------------------------------------ | --------------- | ---------- |
> | Zero-shot                            | 40.9            | 30.2       |
> | RAM-TTT with low-confidence          | 34.1        | 34.5       |
> | **RAM-TTT (high-confidence, ours)**  | **49.5**        | **37.4**   |
>
> We will add this table to the Sec. 4. As shown, using low-confidence pairs severely degrades performance: on MathVista (GPS), accuracy drops to 34.1, which is even worse than the zero-shot 40.9. In contrast, high-confidence examples consistently improve over zero-shot and ICL. This empirical evidence supports our design choice that "reasoning utility” (high LVLM confidence) is a reliable criterion for selecting adaptation examples.
>
>
> **Q2: In stage 2, why do you use the leave-one-out strategy to make $M$ random permutations? For data enhancement?**
>
> **A:** Thank you for the question. The leave-one-out is not for data augmentation, but to instantiate a meta-learning objective at test time.
>
> In Meta TTT, each retrieved example needs to take turns acting as a meta-query, and the other examples form the support set.
> This leave-one-out setup follows the basic idea of meta-learning (e.g., MAML): the model learns to adapt from multiple small "episodes" instead of fitting one fixed support set.
> If we remove the leave-one-out, Meta TTT collapses into standard TTT and loses the meta-learning behavior.
>
> Random permutations address a different issue: the strong order sensitivity of in-context learning in LLMs/LVLMs.
> Prior work has shown that changing the order of examples can change the prediction results, and using multiple orders improves robustness[1,2].
> We therefore construct M random permutations of each support set. As shown in Table 5, increasing M from 1 to 2 improves performance, while larger M brings diminishing or even negative returns, so we set M = 2 as a good trade-off between robustness and efficiency.
>
> [1] Found in the middle: Permutation self-consistency improves listwise ranking in large language models (Tang et al., ACL 2024)
>
>
> [2] Rethinking invariance in in-context learning (Fang et al., ICLR 2025)
>
>
> **Q3: Can you briefly describe the difference between TTT and meta TTT?**
>
> **A:** Thank you for the question. The difference mainly lies in the training objective. Prior TTT methods typically rely on self-supervised objectives constructed from the test query. In our setting, TTT with retrieval fits the retrieved examples using supervised objective. Meta TTT treats retrieved samples alternately as support and meta-query sets under a meta-learning objective.
>
> | Aspect   | Prior TTT                      | TTT with retrieval                | Meta TTT                                         |
> | -------------------------------------- | ----------------------------------------------- | --------------------------------------------- | --------------------------------------------------------------------- |
> | **Training objective**                 | Self-supervised loss from the test query itself | Supervised loss on retrieved support examples | Meta-learning objective using alternating support and meta-query sets |
> | **Applicability to LVLMs Reasoning**   | No  | Yes  | Yes  |
> | **Use of retrieved data**              | Not used                                        | All retrieved examples serve as supervision   | Retrieved examples alternate roles for meta-updates  |
> | **Generalization to final test query** | -  | Weak | Robust     |

---

### Author Response · Authors · 2025-12-03
**Summary of Rebuttal**

Dear AC, SAC, and PC,

We would like to thank you for managing the review process, and also thank all three reviewers for their comments. Below, we briefly summarize the progress made during our rebuttal.

Concretely, to address the reviewers' concerns, we have made the following improvements to our paper:

* **Clearer motivation and contributions**, explicitly framing RAM‑TTT as a unified *retrieve–train–generate* framework, and clarifying the roles of LAR (selecting examples that are both semantically relevant and helpful for LVLM reasoning) and Meta TTT (mitigates overfitting and improves adaptation stability). [6SwX, XVo3, PSrm]
* **Stronger analysis of “reasoning utility” and overfitting**, including a clearer definition of reasoning efficiency and a more detailed analysis of how LAR and Meta TTT improve reasoning and adaptation stability. [6SwX, XVo3, PSrm]
* **Additional experiments**, including: high‑ vs. low‑confidence example selection, robustness to prompt templates, stability under varying adaptation steps, and results on diverse LVLM backbones (Qwen2.5‑VL, Qwen3‑VL, Phi‑3.5‑Vision, Pixtral‑12B). [6SwX, XVo3, PSrm]
* **Implementation and conceptual clarifications**, including the rationale for high‑confidence training pairs, the leave‑one‑out permutation strategy, and a concise comparison between classical TTT, TTT‑with‑retrieval, and Meta TTT. [6SwX]
* **Reporting of accuracy–compute trade‑off**, providing a per‑component latency breakdown of RAM‑TTT and an explicit accuracy–compute comparison against other test‑time scaling methods, showing that RAM‑TTT achieves the best accuracy–compute trade‑off among the evaluated strategies. [XVo3]
* **Clarifications on the primary area and relation to RAG**, explaining that RAM TTT is primarily a transfer learning and meta learning framework with a simple, replaceable retriever. Using ablations over CLIP model sizes and similarity strategies to show that RAM TTT directly benefits from stronger retrievers and is complementary rather than a competitor to existing multimodal RAG pipelines. [PSrm]

Overall, most initial concerns of the reviewers focused on **clarity of motivation**, **depth of analysis**, and **breadth of evaluation**, rather than on fundamental flaws in the RAM‑TTT framework. We believe that our detailed rebuttal and the revised manuscript have now addressed these concerns and provided solid empirical and conceptual support for our claims.

Thank you for your time and consideration.

Sincerely,

Authors

---

### Meta-Review · Area_Chair_UCLS · 2025-12-27

**Summary:**

The paper introduces RAM-TTT, a retrieval-augmented meta test-time training framework that enables LVLMs to adapt to distribution shifts by learning from retrieved examples at inference time. By aligning retrieval with reasoning utility and using meta-adaptation, the method improves multimodal reasoning performance on several benchmarks and models. However, the contribution is diluted by multiple components, the ablation study is found to be unconvincing, and the computational overhead is substantial. Although some presentation and ablation issues were addressed in the rebuttal, the efficiency concern persists: Fig. 6 shows that RAM-TTT incurs roughly a six-fold increase in computation over ICL for marginal accuracy gains.

**Reviewer Concerns:**

The following concerns are addressed:

1.	The ablation study part is not satisfactory (6SwX, PSrm)
2.	Missing mathematical annotations and unclear figure notation (XVo3)
3.	Overfitting robustness claims lack rigorous quantitative validation (XVo3)
4.	Absence of variance reporting, e.g., standard deviations or confidence intervals (XVo3)
5.	Dependence on proprietary prompts with unclear generalisation to unseen prompts (XVo3)
6.	Insufficient comparison with state-of-the-art multimodal RAG methods (PSrm)
7.	Questionable reasoning evaluation due to backbone model limitations (PSrm)

The following concerns are still outstanding:

1.	The core motivation and justification are not clearly presented (6SwX, PSrm)
2.	There is no sufficient evidence to support the improved reasoning efficiency (6SwX, XVo3)
5.	Insufficient justification and explanation for the confidence measure (XVo3)

**Reviewer Scores:**

Reviewer PSrm and XVo3 have claimed to keep the scores. Reviewer 6SwX did not respond but would probably keep the score given the initial positive score optimistically.

---

### Decision · Program_Chairs · 2026-01-26

Reject